# Shaping the topology of folding pathways in mechanical systems

Menachem Stern[1], Viraaj Jayaram[1] & Arvind Murugan [1]

Disordered mechanical systems, when strongly deformed, have complex configuration spaces with multiple stable states and pathways connecting them. The topology of such pathways determines which states are smoothly accessible from any part of configuration space. Controlling this topology would allow us to limit access to undesired states and select desired behaviors in metamaterials. Here, we show that the topology of such pathways, as captured by bifurcation diagrams, can be tuned using imperfections such as stiff hinges in elastic networks and creased thin sheets. We derive Linear Programming-like equations for designing desirable pathway topologies. These ideas are applied to eliminate the exponentially many ways of misfolding self-folding sheets by making some creases stiffer than others. Our approach allows robust folding by entire classes of external folding forces. Finally, we find that the bifurcation diagram makes pathways accessible only at specific folding speeds, enabling speed-dependent selection of different folded states.

[1] Department of Physics and the James Franck Institute, University of Chicago, 929 E 57th Street, Chicago, IL 60637, USA. Correspondence and requests for materials should be addressed to A.M. (email: amurugan@uchicago.edu)

When a heterogeneous mechanical structure like an elastic network or a thin sheet with creases is strained to large extents, it typically shows multiple stable states[1,2]. As we vary the strain level, these states can smoothly deform and appear or disappear in bifurcations, creating a complex network of pathways in configuration space. The geometry and topology of such pathways determines which configurations are smoothly accessible from a given part of configuration space and which ones are not. The response of the material to applied forces is strongly shaped by the network of such pathways[3–5].

Such nonlinear features of configuration spaces have proven to be a double-edged sword. When designed, multiple pathways and multistability can be exploited to create mechanical switches, shapeable sheets, and many other metamaterials[2,6–12]. However, such nonlinear features can also create problems[10,13–15]. For example, self-folding origami, despite the name, has an exponential number of misfolding pathways that meet at a "branch point" at the flat state[13,16–19], making it nearly impossible to fold into the desired folding mode[14,15,20,21]. Similar "branch points" in mechanical linkages pose challenges in robotics and other applications[22–24].

In this work we suggest a new design principle that sculpts the topology of dynamical pathways to desired and undesired states. We focus on elastic networks and creased sheets where rods or plates are connected at flexible joints. We show that heterogeneous stiffness in such joints can completely change the topological connectivity of nonlinear pathways in configuration space. With a distribution of stiffnesses predicted by our equations, undesired pathways can be arranged to end in saddle-node bifurcations. Such bifurcations make undesired states inaccessible from parts of configuration space, at least in the limit of adiabatic folding. Finally, we find that pathways are accessible only at specific folding speeds, allowing dynamical selection between distinct behaviors.

While similar design principles to eliminate dynamical pathways to undesired states are commonplace in protein folding and self-assembly of macromolecular structures and viruses[25–28], such ideas have not been systematically explored in metamaterials design.

Designing the topology of the bifurcation diagram presents several benefits. Once this topology has been designed for a material, it is not modified by entire classes of applied folding forces, but determines the response to such forces. For example, in the context of self-folding origami, other approaches[5] have attempted to find fine-tuned folding forces that will fold a creased sheet successfully. In contrast, our approach produces systems that are truly "self-folding"[29], i.e., our stiffened sheets fold along the desired pathway for almost arbitrary applied forces. Similarly, other approaches[6] have sought to introduce directional asymmetry so that, e.g., individual creases will fold in one way (say, Mountain) but not the other (Valley). Counter-intuitively, our approach shows that even symmetric stiffness in individual creases—an inevitable feature of real materials—can effectively pick a global Mountain–Valley pattern through their collective behavior.

We begin by showing that heterogeneous stiffness in hinges of a mechanical linkage changes the topology of folding pathways by creating saddle-node bifurcations. We show that hinge stiffness predicted from a linear (or quadratic) programming problem can eliminate exponentially many undesired pathways at saddle-node bifurcations and demonstrate such an elimination for folding pathways present at the flat state of thin creased sheets. We show that such a stiffened thin sheet is truly "self-folding" since the sheet can be folded robustly by a host of folding protocols and

forces without any fine-tuning. Finally, we show that controlling the position of saddle-node bifurcations in configuration space, specific folding pathways can be made accessible at specific folding speeds. Consequently, we find that folding speed can select between different target structures.

## Results

**Avoided bifurcation in linkage networks**. We first demonstrate our ideas on a simple but canonical model, namely the 4-bar mechanical linkage[30,31] in Fig. 1a, b. While the structure has only one Maxwell degree of freedom, the flat state is a special point—it sits at a bifurcation where the degree of freedom is branched (and associated with a self-stress mode)[32]. When compressed as shown, the linkage must choose one of the two distinct zero energy motions that conserve rod lengths. The associated energy landscape, at some fixed compression, has two minima corresponding to these motions with an energy barrier (transition state (TS)) between them; see Fig. 1a. (See Supplementary note 1 for precise energy model.) Many studies[3,4,24] have sought to predict and eliminate such "branch points" in complex mechanisms because one of the modes is usually desired and functional, while the other is undesired.

We take a different approach and observe that experimental realizations of such mechanisms[33–35] have imperfections that lift the energies of all the modes. If an imperfection can raise the energy of the undesired zero mode more than it raises the energies of the desired mode and the TS, the undesired mode would disappear in a saddle-node bifurcation with the transitions state.

One such imperfection is stiffness in the joints. We model the stiffness of joint $i$ by a torsional spring of stiffness $\kappa_i$ that is relaxed in the flat configuration shown, i.e., at the branch point. That is, we assume a joint energy $E_i = \kappa_i \rho_i^2/2$, $\rho_i$ being the folding angle measured from the flat state configuration.

We find that if the joints have unequal stiffness $\kappa_i$, the energies of different modes are lifted to different extents. In fact, one of the modes undergoes a saddle-node bifurcation with the TS separating the two modes (see Fig. 1c, d) at a finite folding extent $\rho = \rho_c$ where $\rho \equiv \|\rho\|$. The distance $\rho_c$ is given by a competition between rod compression (or bending in alternative models) at the TS $\sim K\rho^4$, with $K$ a compression modulus, and the spring energy $\sim \kappa\rho^2$; as shown in Supplementary Note 1, $\rho_c \sim \sqrt{\kappa/K}$. Other choices of $\kappa_i$ can eliminate the other mode.

Thus joint stiffnesses change the topological connectivity of undesired modes in state space; see Fig. 1d. As a result, the undesired mode can be made inaccessible from the flat state, which now continuously connects with only the desired mode. If the network is actuated slowly relative to relaxation timescales of the stiff joints, the network will fold into the desired mode and stay in that state even for $\rho > \rho_c$, despite the reappearance of the undesired mode at finite $\rho$.

**Misfolding in self-folding sheets**. Self-folding sheets (or self-folding origami) are structures programmed to have one unique low- or zero-energy mode[6,29,36]. However, self-folding sheets, even when programmed with a single zero-energy mode, have been shown to have exponentially many undesirable misfolding modes accessible from the flat state[14,15]. We show how crease stiffness can change the topological connectivity of these modes and leave only the desired folding mode accessible from the flat state.

To solve the misfolding problem for diverse folding forces, our approach intentionally ignores external folding forces when reprogramming the topological connectivity of modes. Since

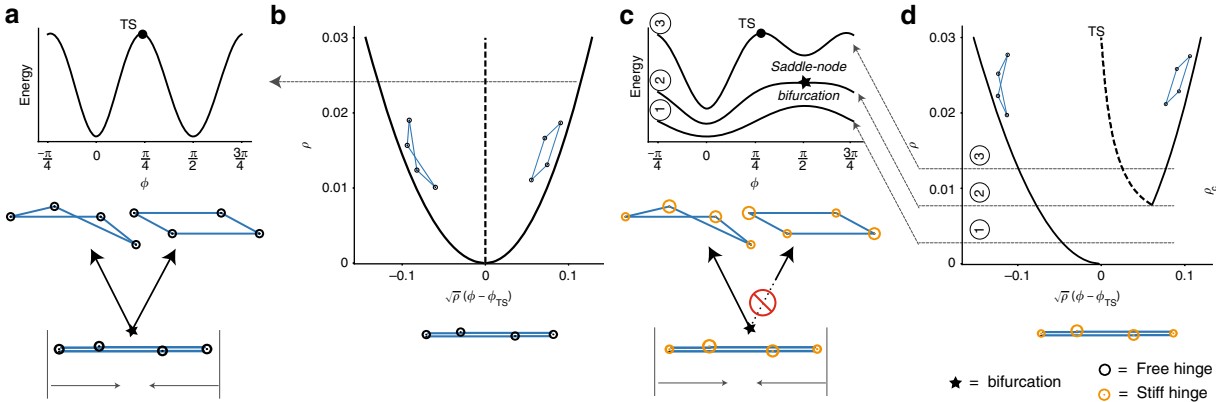

**Fig. 1** Stiff joints in a linkage network can change the connectivity of nonlinear modes in state space. **a**, **b** The 4-bar linkage has only one degree of freedom, but two distinct zero energy motions that meet at a branch point at the flat state, making the mechanism difficult to control. The two motions can be seen as minima in the energy landscape at a fixed total strain $\rho$. **c**, **d** We can eliminate a chosen motion in a saddle-node bifurcation at $\rho_c$ by making the joints stiff to different extents (i.e., adding torsional springs that are relaxed in the flat state; larger orange circles denote stiffer springs). The bifurcation diagram shows that such a stiffness profile changes the connectivity of the two nonlinear modes. One of the two modes is destroyed in a saddle-node bifurcation at $\rho_c$ and is thus inaccessible from the flat state $\rho = 0$ ($\phi$—angle variable in the two-dimensional linearized null space at the flat state, see Supplementary Note 1)

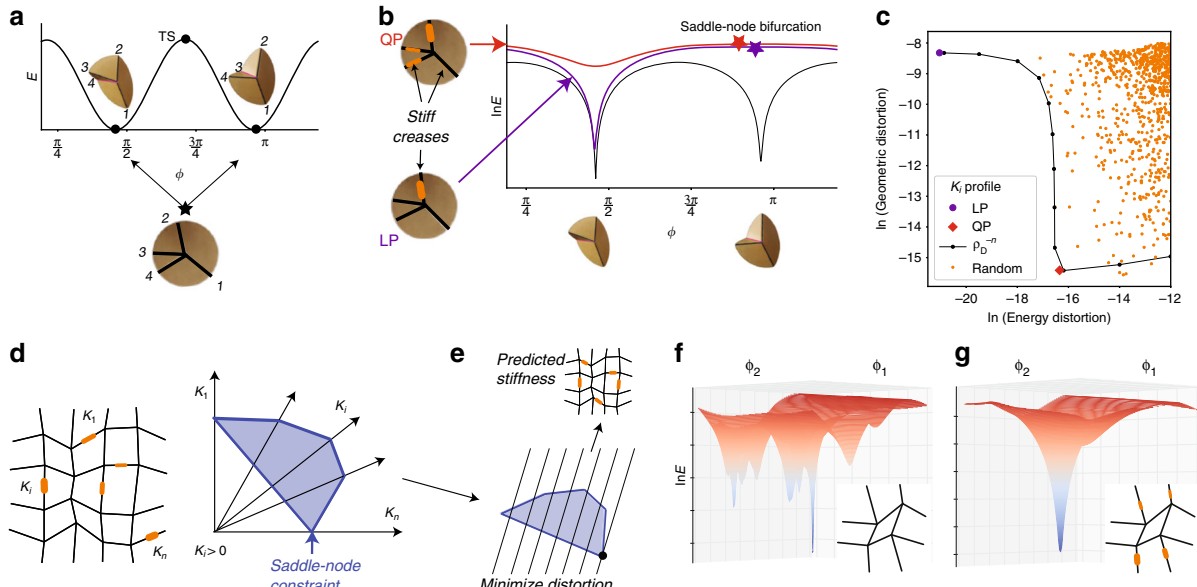

**Fig. 2** Heterogeneous stiff creases can simplify the landscape of self-folding sheets near the flat state. **a** An origami 4-vertex has a choice between two distinct folding modes at the flat state ($\phi$—null space angle variable, see Supplementary Note 2). **b** Stiff creases completely eliminates a chosen mode by combining it with a nearby transition state (TS) in a saddle-node bifurcation (thickness of orange strip indicates stiffness.) **c** Trade-off: stiff creases distort the desired mode while eliminating undesired modes. Stiffness profiles that minimize energy distortion (e.g., the linear programming (LP) method) cause large geometric distortion and vice versa (e.g., the quadratic programming (QP) method). **d** The exponentially many misfolding modes of large sheets are all eliminated if the stiffness profile $\kappa_i$ satisfies a linear constraint, shown here as a simplex. **e** We can minimize distortion (energy or geometry) of the desired mode by optimizing crease stiffness on this simplex. **f**, **g** All but one chosen minimum in a pattern's energy landscape (at small overall folding) can be eliminated by stiff creases predicted by the procedure in (**e**)

folding success relies on the bifurcation diagram topology, our results are mathematically robust to several classes of folding forces as shown later.

**Avoided bifurcation in a 4-vertex**. The atomic unit of self-folding origami is a 4-vertex[37]. Much like the 4-bar linkage, the 4-vertex has one degree of freedom but the flat unfolded 4-vertex is at a branch point, a meeting point of two distinct folding

motions[2,5], distinguished by the Mountain–Valley states of the creases (Fig. 2a)[38,39]. These two motions are shown as zero energy minima in Fig. 2a using a model of vertex energy presented in Supplementary Note 2, with a TS separating them. This binary choice is the origin of the exponentially many misfolds of large self-folding sheets.

As with the 4-bar linkage, we wish to lift and eliminate one of the two folding motions, making it inaccessible from the flat state. We

introduce stiffness at the creases, an inevitable feature of most material implementations. We model such stiffness of crease $i$, as a torsional spring with $\rho = 0$ rest angle and energy $E_{\text{Crease}, i} = \kappa_i \rho_i^2 / 2$. The energy of the origami vertex is,

$$E = E_{\text{Vertex}} + E_{\text{Crease}}, \qquad (1)$$

where $E_{\text{Vertex}}$ accounts for bending of vertex faces[37] and $E_{\text{Crease}} = \sum_i \kappa_i \rho_i^2 / 2$ accounts for crease stiffness. For details on the energy model see Supplementary Note 2. Crucially, $E_{\text{Vertex}}$ scales with a high power $\rho^4$ for the two special folding motions.

Let us find the conditions on $\kappa_i$ for lifting and eliminating a chosen mode—the "undesired mode"—of the 4-vertex. We assume the folding angles of the undesired mode and the desired mode are $\tilde{\rho}_U$ and $\tilde{\rho}_D$, respectively and that of the TS separating them is $\tilde{\rho}_{TS}$, all assumed to be defined near the flat state and normalized (with unity magnitude); see Fig. 2a.

Let $E_{TS}(\rho)$ be the energy of TS at some chosen total folding $\rho \equiv \|\rho\|$. As the vertex null space (at fixed $\rho$) is one-dimensional and compact[14], these features ($\tilde{\rho}_U$, $\tilde{\rho}_D$, $\tilde{\rho}_{TS}$, and $E_{TS}$) can all be computed numerically efficiently using peak analysis. Here, we will focus on eliminating the undesired minimum up to a distance $\rho_c$ from the flat state and return to larger folding behaviors later. To lift and eliminate the undesired minimum, we should choose a heterogeneous stiffness profile that raises the undesired mode more than the TS. This constraint—requiring a saddle-node bifurcation—can be written as,

$$\frac{1}{2}\rho_c^2 \sum_{i \in \text{creases}} \kappa_i \left[ (\tilde{\rho}_U)_i^2 - (\tilde{\rho}_{TS})_i^2 \right] \geq E_{TS}. \qquad (2)$$

In addition, all crease stiffnesses must be nonnegative:

$$\kappa_i \geq 0. \qquad (3)$$

Note that both constraints are linear in the stiffnesses $\kappa_i$.

Any set $\kappa_i$ satisfying the above constraints that predominantly raises the undesired mode will eliminate it will eliminate the undesired mode in a saddle-node bifurcation at a total folding distance $\rho_c$, making it inaccessible from the flat state.

Only the desired mode is stable in the neighborhood of the flat state but it can be significantly distorted by the stiff creases. As shown in Fig. 2, with stiff creases, the desired mode is of nonzero energy ("Energy distortion") and can also have distorted folding angles ("Geometric distortion", defined by one minus the normalized dot product of the desired mode and the obtained minimum). We wish to formulate design principles for choosing stiffness profiles $\kappa_i$, consistent with the above constraints, that best facilitate designed folding motions.

We devise two design strategies: (1) minimizing energy of the desired mode (energy optimization) and (2) minimizing geometric distortion of the desired mode (geometric optimization). We find that different crease stiffness profiles generally trade-off energy and geometric distortion.

Energy optimization is simple: the desired mode has non-zero energy $E(\rho_D) = \sum \kappa_i (\rho_D)_i^2 / 2$ because of crease stiffness. As this function is linear in $\kappa_i$, optimization subject to the saddle-node constraints equations (2) and (3) is equivalent to a Linear Programming (LP) problem[40,41]:

$$
\begin{aligned}
\underset{\kappa_i}{\text{minimize}} \quad & E(\rho_D) = \frac{1}{2}\sum_i (\rho_D)_i^2 \kappa_i \\
\text{subject to} \quad & \rho_c^2 \sum_i \left[ (\tilde{\rho}_U)_i^2 - (\tilde{\rho}_{TS})_i^2 \right] \kappa_i \geq 2E_{TS}, \\
& \kappa_i \geq 0, \ i \in \text{creases}.
\end{aligned} \qquad (4)
$$

LP problems are solved in polynomial time, as long as an efficient algorithm is used. Further, the optimal stiffness profile $\kappa_i$ is generically guaranteed to be sparse. In a 4-vertex, only one crease needs to be stiff.

Geometric distortion is minimized if fold angles in the surviving minimum with stiff creases closely corresponds to the fold angles $\rho_D$ of the desired mode without stiff creases. Here, we use the gradient of the energy with stiff creases, but evaluated at $\rho_D$, as a proxy for such geometric distortion. As shown in Supplementary Note 4, this proxy, after projecting out the component of the gradient in the $\rho_D$ direction, is $F_{QP} = \rho_D^2 \sum_{i \in \text{creases}} \kappa_i^2 (\rho_D)_i^2 - \sum_{i,j} \kappa_i \kappa_j (\rho_D)_i^2 (\rho_D)_j^2$. $F_{QP}$ is a positive semidefinite quadratic function of $\kappa_i$. Optimization of $F_{QP}$—with the linear constraints in Eqs. (2) and (3)—is facilitated by efficient Quadratic Programming (QP) algorithms.

In practice, the LP and QP prescriptions do well at optimizing their respective strategies (i.e., energy and geometry) for a single vertex. Figure 2c shows how these prescriptions indeed do better than choosing random stiffness profiles that satisfy the constraints. The black line $\kappa_i \sim (\rho_D)_i^{-n}$ for positive $n$, shows that stiffness profiles trade-off energetic and geometric distortion.

**Stiffness profiles in large self-folding sheets.** Large origami patterns have exponentially many distractor minima states, making them near impossible to fold correctly[14,15]. Still, the ideas of the previous section can be used to lift all but one of these minima at small folding angles. Crucially, the desired self-folding motion of a large pattern[35,42] is consistent with exactly one of the two folding modes for each of its constituent 4-vertices. Thus, for a pattern with $V$ vertices, the saddle-node constraint in Eq. (2) generalizes to $V$ linear constraints, one for each vertex $v$:

$$\rho_c^2 \sum_{i \in \text{creases of } v} \kappa_i \left[ (\tilde{\rho}_{U,v})_i^2 - (\tilde{\rho}_{TS,v})_i^2 \right] \geq 2E_{TS,v}. \qquad (5)$$

Note that the constraints are dependent since vertices share creases. These linear constraints, along with $\kappa_i > 0$, define a simplex in the space of crease stiffnesses as shown in Fig. 2d. We can still use LP and QP algorithms as before to find optimized stiffness profiles.

**Larger folding angles and adiabatic folding.** Figure 2f, g shows that applying a LP stiffness profile to a quad pattern lifts all but one minimum close to the flat state. Sampling many large patterns shows that LP and QP indeed optimize their respective strategies However, while folding the quad with stiff creases in Fig. 2g retrieved the desired structure, we noticed that folding beyond a certain angle gives rise to many new minima (Fig. 3a). To understand this, note that the saddle-node bifurcation constraint, Eq. (2), only ensures the absence of undesired modes up to a total folding $\rho_c$ at which $E_{TS}(\rho_c)$ is computed. Intuitively, crease stiffness ($\sim \rho^2$) becomes less important than face bending ($\sim \rho^4$) as folding proceeds and undesired modes are restored in a series of saddle-node bifurcations.

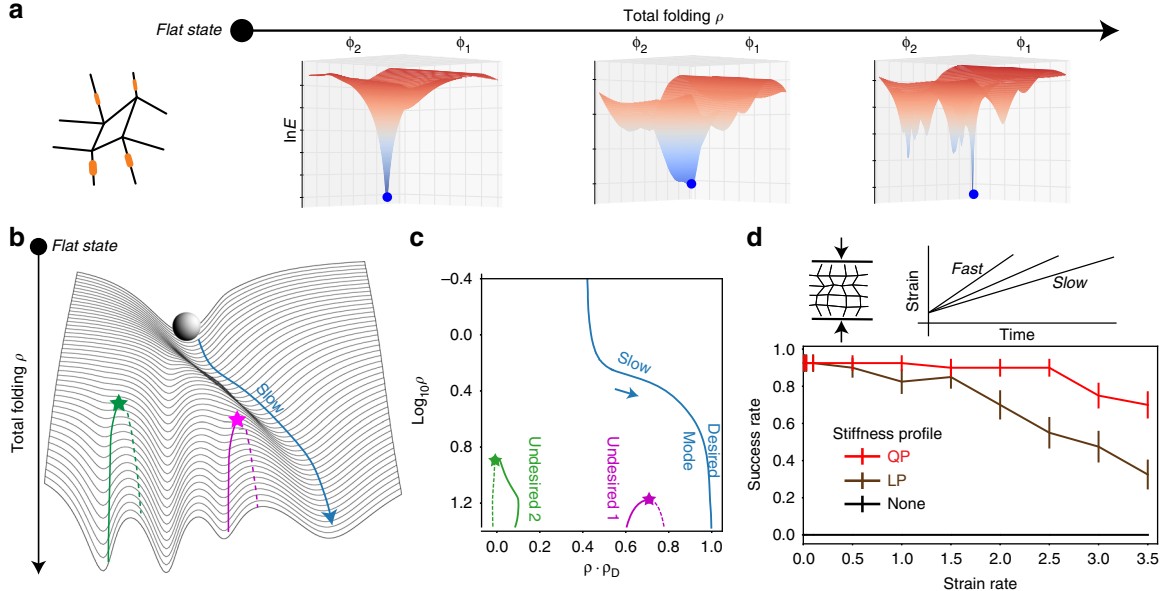

**Fig. 3** Stiff creases change the topological connectivity of undesired modes and promote folding at slow speeds. **a** The energy landscape of a quad with stiff creases has a unique mode at low strain $\rho$, but becomes more complex at higher strain. **b** However, slow folding will recover the desired state provided the unique mode at low strain is continuously connected to the desired state and does not undergo a saddle-node bifurcation (blue line). Slow folding can be successful even if the unique mode at small strain $\rho$ is quite distorted relative to the desired mode. **c** Bifurcation diagram as a function of total folding $\rho$ for a specific 16-vertex pattern shows select undesired modes (solid lines) being eliminated at bifurcations with saddles (dashed lines). Only the desired mode (blue) survives—albeit distorted—to the flat state. **d** Simulations of folding at a finite strain rate (relative to relaxation timescale of hinges) show high success for slow folding and failures at higher folding speeds (Data from 50 random 16−vertex patterns)

At first sight, the reappearance of undesired modes at large $\rho$ might seem disappointing. However, if folding is carried out adiabatically—i.e., slowly relative to hinge relaxation timescales—these modes do not impact folding at all. Adiabatic folding, by definition, will follow the continuous deformation of the unique low-$\rho$ minimum (blue paths in Fig. 3b, c), even if it is significantly distorted relative to the desired state. Thus, for successful adiabatic folding, the only condition is that the unique low-$\rho$ mode is continuously deformed to the desired structure at large $\rho$ (Fig. 3b). Figure 3(c) show the bifurcation diagram for a 16-vertex pattern with stiff creases predicted by LP. The unique low-$\rho$ mode is significantly distorted relative to the desired state (i.e., has low-dot product). Nevertheless, this mode is continuously deformed to the desired state along the blue path, which was followed in slow folding simulations. Undesired states, on the other hand, are not continuously connected to the low-$\rho$ mode.

To test whether our stiff crease prescriptions are able to consistently create such adiabatic pathways, we sampled 50 random patterns, each with a programmed low-energy motion using the loop equations of ref.[35,42]. Such patterns are expected to have ~$10^3$ higher energy undesired modes[14], corresponding to motions that jam close to the flat state. Accordingly, we almost never succeed in folding in the desired low-energy mode with generic folding torques. and thus folding almost always fails (Fig. 3d).

We then augmented the sampled patterns with stiff creases resulting from LP and QP prescriptions and simulated folding at varying speeds. In simulations, we assume the crease hinges follow a first-order equation with a relaxation timescale $\tau_{relax}$; this timescale is known to vary with material implementation[43]. These stiff patterns achieve a success rate in excess of 90% when folded slowly (Fig. 3d), compared with the expected <0.1% success rate with free folding creases. Thus our stiffness heuristics are useful for slow folding, yet imperfect.

The small fraction of failed cases represent patterns where the unique low-$\rho$ mode and the desired high-$\rho$ mode undergo distinct saddle-node bifurcations at intermediate $\rho$ and thus do not connect up. Such bifurcations are mathematically forbidden if these states are the lowest energy states for all $\rho$. Complex optimization methods that account for details of nonlinear energy landscape at all intermediate $\rho$ might be able to better protect from such bifurcations. However, we find that simple heuristics, e.g., based on the energy of low- and high-$\rho$ states alone, are sufficient to protect the adiabatic pathway from bifurcations for complex patterns. See Supplementary Note 5 for more analysis of failures.

**External folding forces applied to creases.** Our crease stiffness prescriptions are meant to eliminate undesired modes in the intrinsic energy landscape of a sheet and not just for a particular model of folding—hence no particular folding forces are assumed. Once undesired modes are eliminated, many typical classes of folding forces cannot reintroduce such modes near the flat state.

Besides strain-controlled folding tested above, another important class of folding forces[6,44] involves folding torques $F_i$ applied to specific creases $i$; see Fig. 4. A related method involves setting target folding angles $\rho_i^{target}$ for particular creases (see Supplementary Note 3). Near the flat state, both of these methods change the energy landscape by a linear tilt ($\sim F_i\rho_i$). Mathematically, such tilts cannot create new undesired minima close enough to the flat state. We tested folding success in these methods of actuation as a function of the number of actuated creases. Folding success is enhanced by orders of magnitude due to the stiff creases predicted here as shown in Fig. 4. Hence, our approach to modifying the topology of the bifurcation diagram is also useful when external folding forces are present—in fact, such a modification is necessary for successful folding.

Earlier works[5] have tried to find such specific folding torques or folding springs to fold along a desired mode. Mathematically,

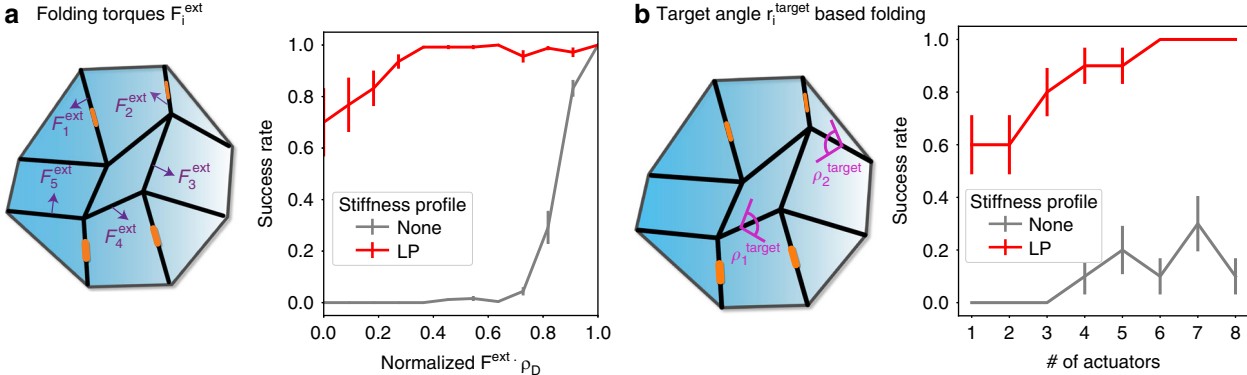

**Fig. 4** Sheets with stiff creases dramatically improve folding for a wide range of external forces applied to specific creases. We consider folding by (**a**) different folding torques $F_i^{ext}$ or (**b**) springs that target desired fold angles $\rho_i^{target}$ for different creases $i$. **a** Forces $F_i^{ext}$ are much more likely to fold into the desired mode with stiff creases (red data) (predicted by linear programming) than with freely folding creases (gray). Even folding forces $F_i^{ext}$ very poorly aligned with the desired pathway $(\rho_D)_i$ result in the desired pathway (averaged over 5 random 16-vertex patterns, 100 random $F_i^{ext}$ for each data point). **b** Patterns folded using springs of given target angle $\rho_i^{target}$ on select random creases $i$. Successful folding into the desired mode is dramatically improved in patterns with stiff (LP predicted) creases compared to free folding creases. Even a single actuator is successful a significant fraction of the time (data averaged over ten random 16-vertex patterns)

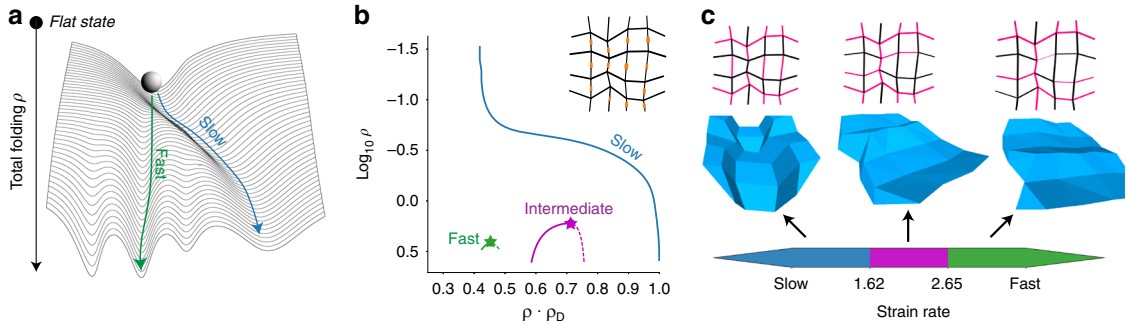

**Fig. 5** Folding speed can controllably select between different folding pathways. **a** While slow folding follows the continuous deformation of the unique mode at low strain $\rho$ (blue), fast folding results in a state that most "resembles" that low-$\rho$ mode (green). If the unique low-$\rho$ mode is significantly distorted in geometry relative to the slow folding target, slow and fast folding can result in very different outcomes. **b, c** We systematically attempted folding at different strain rates (relative to a fixed hinge relaxation timescale) for the 16-vertex pattern with stiff creases shown. We find three distinct outcomes at slow, intermediate, fast rates that completely differ in their Mountain-valley states, geometry and energy. The slow folding outcome corresponds to following the blue path in (b) while the intermediate and fast pathways cross over from blue to the magenta and green modes, respectively, at some intermediate folding angles

such approaches appear similar to ours since they both involve quadratic modifications to the energy function. However, our crease stiffness is a quadratic potential centered at the branch point (i.e., the flat state), and hence is able to modify the topology of that point successfully while springs with finite target angles[5] are effectively linear tilts at the branch point. As noted in ref. 5, successfully folding the pattern using such a method requires undesired branches to have negative dot products with the desired branch, a very unlikely scenario for larger patterns. In contrast, our quadratic potentials at the flat state face no such restriction and thus work in a dramatically larger context.

Our approach is also different in practice. Prior approaches[5,14] sought sheets with freely folding creases that must be carefully actuated using calculated folding forces. Our approach designs sheets with calculated crease stiffness profiles that can be carelessly actuated.

**Folding speed-dependent target structures**. We have seen that the unique low-$\rho$ minimum funnels adiabatic folding to the desired state in a glassy landscape, even if the unique low-$\rho$ mode is significantly distorted relative to it. However, the success rate drops with folding rate; see Fig. 3d.

Such a drop in success rate is to be expected since very fast folding essentially takes the pattern from the unique low-$\rho$ state to high-$\rho$ state with quenched geometry and then relaxes to the nearest minimum. Thus, as suggested by Fig. 5a, fast folding from the unique low-$\rho$ minimum reproducibly picks the folded configuration with closest geometric resemblance to the low-$\rho$ minimum.

These considerations suggest an intriguing possibility—programming the bifurcation diagram using stiff creases can program different folding pathways that are followed at different speeds of folding.

We tested this hypothesis on a 16-vertex pattern with LP springs whose unique low-$\rho$ mode has significant geometric distortion relative to the adiabatic folding outcome; see Fig. 5b. We systematically folded this structure at increasing speeds relative to its hinge relaxation timescale. We find three completely distinct but reproducible folded structures in regimes of slow, intermediate and fast folding; see Fig. 5c.

## Discussion

In this paper, we argued that metamaterials design should be conceptualized as targeted design of an entire dynamic pathway that avoids undesired behaviors, and not just target a desired final state. We showed how such pathways and their topological connectivity can be programmed by controlling the bifurcation diagram; we applied our method to remove the exponentially many misfolding motions intrinsic to self-folding origami.

We showed that the bifurcation diagram can be modified by stiff joints, an inevitable feature of most experimental realizations of origami, linkage networks, and other metamaterials. Thus, our proposal is conservative—it does not require specific directional information at hinges[6], temporal staging[45], or using nonflat sheets[37]. Our general approach applies to any other heterogeneous bulk imperfection that couples to different folding modes unequally.

A particularly intriguing direction suggested by our work is the ability to geometrically program different behaviors at different speeds. These outcomes can have independently tuneable mechanical properties, such as energy absorption[35]. While such complex speed-dependent phenomena are actively studied in materials (e.g., cornstarch[46,47]), our approach suggests that speed-dependent behaviors can be dictated simply by the bifurcation geometry of the metamaterial. A recent study[48] independently demonstrates how heterogeneous stiffness can shape multi-stage folding pathways, allowing robust and predictable folding of metamaterials.

**Code availability**. MATLAB code to compute LP and QP stiffness profiles is given as Supplementary Software.

## Data availability

Data supporting the findings of this study are available from the corresponding author on request.

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

## Acknowledgments

We thank Alfred Crosby, Heinrich Jaeger, Sidney Nagel, Jiwoong Park, and Thomas Witten for insightful discussions. We acknowledge NSF-MRSEC 1420709 for funding and the University of Chicago Research Computing Center for computing resources.

## Author contributions

M.S., V.J., and A.M. developed the theoretical tools, carried out simulations, analysis, and wrote the manuscript.

## Additional information

**Competing interests:** The authors declare no competing interests.

