## [Peer Review File · Nature Communications]

Reviewers' comments:

Reviewer #1 (Remarks to the Author):

This is an extremely interesting paper about how to design origami (or more generally linkages of bars, but the focus is on origami) to select a particular folding pathway so it reaches the desired target state under a wide range of forces. Some recent work by different groups including the authors' has established that starting from a flat state, there are usually a great many different pathways that a creased paper (with no energy on its creases) can follow, and since all but one of these is the pathway one wants to follow, it should take a highly controlled, precise force to be able to follow this pathway. This is true even if one designs the origami to have a one-degree-of-freedom folding pathway, as the flat state is a highly singular state where many different one-degree-of-freedom pathways intersect, including the designed pathway and lots of undesired pathways. So the goal of designing self-folding origami (i.e. origami which folds spontaneously under thermal fluctuations, or else under a wide range of external forcings) might seem hopeless. However this paper offers an intriguing and in hindsight quite simple idea: make the crease stiff, and with different stiffnesses, and then the energy will change by different amounts along different pathways and so presumably only one pathway will be the lowest in energy initially, and this is the pathway that will be followed when external forcing is applied.

This idea seems so simple, and yet the authors argue quite convincingly that it is effective: they show that one can solve for the required stiffnesses by solving a linear programming problem, hence efficiently, and they demonstrate through simulations under several different kinds of external forcings that this strategy frequently allows the system to follow the desired pathway and end up at the desired folded state. In addition they argue that the number of creases that must be made stiff, is usually much smaller than the number that must be activated if the only control one has is in applying forcings to creases.

I think this is an excellent paper, with an original idea proposed to solve an important problem — not only because self-folding origami is a promising way to manufacture materials, both on the macroscale and possibly also the nanoscale — but also because the problem of designing assembly pathways occurs in a wide range of materials and statistical mechanics problems, where a system does not always have time to find its ground state so the pathways it follows can determine what it assembles into— while the problems take different forms in different contexts, there may still be a link between the strategies used in design. The work is competently and thoroughly carried out, and speculates on the wider implications of the work, making links for example from their observation that different rates of strain choose different pathways, to non-Newtonian materials which have strain-dependent rheologies, an idea I found intriguing. I would recommend publication after some small changes. I do have some small questions/concerns listed below but they are for the most part relatively minor.

— Please proof-read — while the paper is overall well-written there are a few blatant typos.

— How does one know there is a solution to the linear programming problem setting the creases (Eq 6)? And if there is a solution, how does one know that there is some tolerance around this solution, so that if one doesn't manufacture exactly the stiffnesses one predicts are optimal, the structure can still fold along the desired pathway? Or in other words, how does one know that the volume of possible solutions in kappa-space is large enough? If the number of undesired pathways increases exponentially with the number of vertices, then isn't it possible that the volume of solutions for a single pathway in kappa-space decreases in a similarly exponential fashion?

This concern would be important to address to make sure the strategy the authors suggest is actually feasible in practice.

There may be a simple argument to argue that there is actually quite a lot of tolerance on the

creases — for example the number of edges (E) is about twice the number of vertices (V), so if there are roughly 2^V pathways and roughly 2^E possible solutions in κ -space (if to a rough approximation the linear programming solution selects a subset of creases to activate), then there are more than enough choices of edges to cover the pathways with some tolerance. However, I'd be interested in what the authors think.

— How does one know what the transition state is (eg in Eq 2)? And for a material with lots of creases, there should be many transition states — they all appear in the minimization problem (Eq 6), so presumably must all be found — how does one find them? This sounds like a hard problem.

— I didn't understand how Eq 2 ensures that the energy of $\tilde{\rho}_D$ is lower than both the transition state and the undesired mode. Eq 2 seems to be an equation which says that the energy of the undesired mode, is greater than the energy of the transition state. But says nothing about the desired mode. Maybe I missed something? Perhaps the authors could clarify, either in their response, or the paper?

— In Fig 2C there is a red point which appears to have lower Distortion than any others (in the south-west corner). How is this possible, if the filled-in red dot was found by optimizing over distortion?

- SI Fig 2 — can you say something about the color scheme?

Reviewer #2 (Remarks to the Author):

Review of NCOMMS-18-17007

Shaping dynamical pathways in mechanical systems

The paper under review considers a pitfall of self-folding mechanisms, framed in the context of mechanical metamaterials, whereby such a mechanism could fold into an undesirable state. That is, the configuration space of such a mechanism may have branch points, and in order to control the mechanical behavior, one wishes to guarantee movement along the desired path through such branch points. In this paper, we see a new method for such mechanical control via the utilization of intentional imperfections in the folding joints, e.g., making some joints more stiff than others. The authors describe how such imperfections can be used to tune the mechanics and thus avoid undesirable folded states.

This work is quite excellent. The authors' analysis of varying stiffness of folding joints is complete and, to my eyes, correct. They consider the 1-dimensional linkage case as well as 2-dimensional folded sheets. While they restrict themselves to 4-bar linkages in the 1D case and degree-4 vertices in the 2D case, these cases are the most immediately useful to current engineering applications, and since such systems are 1-DOF, they make the most sense to consider initially. Aside from some comments on the paper's exposition (see below), I see no flaws in this work.

This paper is certainly of great interest to researchers in the field of folding mechanics. While it does serve as an interesting lesson of what can happen in metamaterial design, it really is specifically about folding- or origami-inspired metamaterials. Any researchers who have been following the surge of interest in folding in the physics and engineering communities will find this paper of significant value. It will definitely make a positive change/impact on the field.

After some thought, I am deciding to recommend this paper for publication in Nature Communications. However, I want to explain my hesitation, because I think it suggests a slight

change in emphasis by the authors.

When I first read this paper, I was intrigued but kept thinking to myself, "This is not surprising at all." The idea that adding stiffness to creases would alter the configuration space, especially if one incorporates the total strain into the configuration space, is completely unsurprising. Thus, while the work in this paper is very complete, solid, and technically new, I initially thought that I might not recommend it for publication because the over-all conclusion seemed somewhat obvious.

But further examination of the results changed my mind for the following reason: The addition of stiffness to the folding joints can be used to tune the configuration space even if different folding torques are applied to the joints. This result is, I think, not emphasized in the paper; it is only mentioned in the short "Other models of folding forces" section on pages 5-6. I will comment on this in some more detail below, but the point I want to make is that this aspect of the authors' stiffness approach is, to me, quite surprising and useful. That is, one could argue that adding joint stiffness is tantamount to using different torques at the joints (which has been studied previously), that these two different approaches might be, at least mathematically, equivalent. E.g., one could model each with a vector field on the configuration space of the folding angles ρ_i . However, the results of the simulations shown in the SI Figure 4 show a very high success rate of desired self-folding over a variety of joint torques with stiffness added versus low success rate without the added stiffness.

Therefore I strongly suggest that the authors call more attention to this aspect of their results. I think that engineers in particular will find it interesting that adding joint stiffness seems to give a much better rate of successful self-folding than programming specific torques for the joint actuators.

Therefore I do recommend this paper for publication in Nature Communications.

Below are more specific comments and suggestions for the authors.

(1) The introductory paragraphs of the paper do not do a good job of preparing the reader for the main topic of the paper. That is, nowhere is it specifically stated in the first 4-5 paragraphs that the authors will be studying folding processes. In fact, the first indication that the subject of the paper is folding mechanisms is at the end of the 4th paragraph when suddenly "joint stiffness" is mentioned --- nowhere before is it stated that the materials being considered are folding along joints or crease lines. Even the title of the paper does not suggest this. (The abstract does, but the main body of the text should be written to be understood independently of the abstract.) I understand the desire to highlight the general connection to mechanical metamaterials, but the paper should state up-front that the main subject is folding.

(2) A minor gripe about notation: The variable ρ is used for the total strain of the system, and at the same time other variants of ρ are used for other variables. E.g., in the SI, $\vec{\rho}$ is the vector of coordinates for points of a 4-bar linkage, and ρ_i is the folding angle of crease i in a 2D folded crease pattern. This is a bit confusing. Maybe a different variable could be used for the total strain?

(3) I remain confused as to what the variable ϕ represents. The SI describes it (in the 4-bar linkage and degree-4 vertex cases) as the "mixing angle of the two zero-modes spanning the 2-dimensional null space." What is meant by "mixing angle"?

(4) The first page of the SI seems to have some typos. In the sentences after equation (1) we see:

$$\vec{\rho}_0 = [x_B=L_1, y_B=0, x_C=L_1+L_2, y_B=0]$$

Shouldn't the last term there be " $y_C=0$ "?

And later in the same paragraph the authors write, "To see this, note that to lowest order in ρ ,

the two zero-energy motions are..." Should that be $\vec{\rho}$? From the main text of the paper, I am still assuming ρ is the total strain, yes? That doesn't seem to be what you mean here, or is it?

In the next paragraph is mentioned ρ^4 . Is this also meant to be the total strain? Or should it be some part of $\vec{\rho}$? Or maybe $|\rho|^4$? This notation is also present elsewhere, such as in the caption to SI Figure 1 and 2.

(5) Perhaps I am missing something, but I am surprised that the relative lengths of the creases in the 2D case (Section B in the main paper) do not seem to be taken into consideration. Adding stiffness to a longer crease versus a shorter crease should have a different impact to the energy landscape, yes? Perhaps the relative lengths of the creases are captured in the energy model, but if so, I don't see how this is done. Perhaps a clarifying sentence addressing this is needed somewhere.

(6) In the SI, Section III.A., when describing their simulations of torque-based folding, the authors state, "It is hard to call such high dot-based folding 'self-folding' since such actuation requires a large number of actuator creases and the torques F_i on each crease needs to be tuned carefully." However, the authors seem not to be considering the approach in the paper:

Tomohiro Tachi and Thomas C Hull, Self-Foldability of rigid origami, *Journal of Mechanisms and Robotics*, 9 (2), 2017, 021008-021017.

(which note is not the same paper as the authors' reference [35]). In that paper an approach to actuator-based self-folding is made whereby the torques are tuned to make low (or, if possible, zero) dot product with the undesirable configuration space branches. This greatly reduces the chance that the undesirable branches will be followed. The authors' point may remain the same, but perhaps programming torques to have low dot product with the tangents of undesirable branches would have a better success rate in the absence of added joint stiffness.

(7) In the references of the main paper:

* In reference [2], the journal Science needs to be capitalized.

* reference [20] seems to be a duplicate.

Reviewer #3 (Remarks to the Author):

Rigid-foldable origami patterns made of quadrilateral faces and valence-four vertices generically have one degree of freedom: actuating one crease edge folds the entire pattern. The same is not true when the pattern is in the flat configuration: multiple zero-energy folding modes exist in the flat state, so that in practice origami must be pre-creased with mountain and valley folds to bias the pattern towards the desired folding mode. This paper studies ways of eliminating undesired folding modes by adding torsion springs to a sparse set of crease edges; if chosen carefully, these springs modify the energy landscape in a small neighborhood of the flat state so that there is a unique elastic-energy-minimizing mode, guaranteeing that the pattern, when quasistatically folded, will fold in that mode. The paper presents two algorithms for selecting springs to add: one minimizes the energy (which, in the ideal case, is zero) of the desired folding mode, while prohibiting all others; the second attempts to minimize the geometric distance between the desired folding mode of the unmodified origami pattern, and the perturbed configuration that would actually be seen when folding the pattern with added springs (these springs introduce additional internal forces that will be balanced by strain at the faces and edges of the origami pattern during folding). The former leads to a linear programming problem, and the latter, a quadratic programming problem, both of which can be easily solved using numerical techniques. The paper demonstrates that the optimized springs achieve the goal of minimizing energy and geometry distortion far better than randomly placing springs.

Overall, I agree that the problem of taming the complex and degenerate energy landscape of origami patterns is an interesting and important problem whose solution would greatly improve the practical usefulness of folding structures to engineers and materials scientists, by making the structures more controllable. The approach suggested by the paper, though simple, seems reasonable and was validated on some large origami patterns. That said, I do have some technical questions and concerns:

1) Equation 2 is the heart of the paper, as it is used in both the LP and QP optimization algorithms to try to ensure that for small strains, only one folding mode away from the flat state is in static equilibrium. But equation 2 doesn't quite achieve this goal: adding springs does increase the energy of the initially-zero-energy undesirable configuration $\tilde{\rho}_U$, but also shifts the energy minimizer $\tilde{\rho}_U^*$ away from $\tilde{\rho}_U$: the energy of $\tilde{\rho}_U^*$ could be less than $E_{\{TS\}}$ even though $\tilde{\rho}_U$ has greater energy than $E_{\{TS\}}$. See for instance SI figure 3B and 3C, where it is evident that the location of the energy minima on the ϕ -axis changes as ρ increases. Perhaps a "safety factor" should be added to the right-hand side of equation 2; it would be even more satisfying if this shift in minimizer were accounted for theoretically.

2) Geometric distortion is quantified (in the QP optimization) by looking at the energy gradient at the desired (and, initially, zero-gradient) folded configuration. Why is the gradient a good surrogate for geometric distortion? This choice seems dubious to me: it essentially assumes that the Hessian of energy does not change much near ρ_D (reasonable) and that the Hessian is isotropic there (seems unlikely). Were any experiments performed to measure how closely the gradient magnitude correlates to true distance (in the Hausdorff sense, say) between the unaugmented and spring-augmented desired folding modes of the pattern? Are there any theoretical reasons to believe that the gradient is a good measure of geometric distortion?

In terms of exposition, the paper is mostly clear, albeit only when interleaving reading the main paper with the SI. I wish there had been more results about folding-rate-dependent structures (I.C), as this is a very intriguing yet underexplored mechanism for controlling deployable structures, and less space spent on the four-linkage example, which it is not substantially simpler to understand than the four-crease origami unit cell of section I.B. Some detailed comments:

- the paper correctly points out that the proposed method relies on bifurcations existing only at the flat state. For rigid-foldable flat origami this assumption seems reasonable, but it would be useful to have more discussion about when, in practice, this assumption is likely to be invalid.
- More explanation of figures 1B and 1D would be useful. What is $\phi_{\{TS\}}$? Why do the left panels of 1B and 1D use a different vertical axis than the right panels (energy cross-sections)? Why are some dots in 1C bigger than others (this is explained in the SI but not, I believe, in the main text or figure caption.)
- It is stated that $\rho = \|\vec{\rho}\|$ in the top-right of page 2, but ϕ is used heavily beforehand (including in Figure 1 and the SI) and it would have been useful to have this definition earlier.
- "our approach intentionally ignores specific folding forces": that is to say, the forces driving folding, not the elastic forces arising from folding.
- the main text calls an elastic formulation that models rod stretching the "alternative model," but in the SI it is the bending model that is "alternate."
- speaking of the bending model, it is not spelled out in detail (presumably the rods are now assumed exactly inextensible, but can deflect (with what profile?) between the endpoints?) and it is not clear to me that the analysis is identical to the case of Hookean unbending rods.
- The two bending modes in 2A appear to be the same mode (with different sign); this may be an artifact of the rendering but it would be useful if these two modes were more clearly different.
- In figure 2C, how is geometric distortion measured? Page 5 mentions having "low dot product"

but that does not clarify for me.

- I don't understand the frontier of black dots and lines in figure 2C. What is ϕ^{-n} ?
- A sentence or so explaining "the vertex constraint" (page 3) in the main text would be useful.
- What does it mean for $\tilde{\rho}_U$, etc to be "normalized"?
- The notation in equation 4 is rather sloppy; presumably bold-face ρ , squared, signifies component-wise squaring of all entries of $\vec{\rho}$. I'd prefer notation like \odot or \star for component-wise multiplication, with a few words of explanation.
- "the optimal stiffness profile is guaranteed to be sparse": why is this?
- What is a "radial component" of the gradient (equation 5)?
- "Equation 2" -> "Equation 2"
- Section 1 in the SI interchangeable treats the linkage as having one degree of freedom, with three rigid kinematic constraints; having four degrees of freedom, with two points A and D pinned; and having the full eight degrees of freedom (in equation 2, eg). It would be less confusing to make the parameterization of the problem more consistent and explicit.
- " $L_{AB} \cdot L_{CD}$ ": comma?
- Why is energy quartic in ρ^4 , close to the flat state? I agree with the surrounding sentences (that energy due to extension vanishes to quadratic order, etc) but I do not see how it follows that the cubic term vanishes.
- The color scheme in SI figure 2 is confusing (especially in black and white): energy is everywhere-positive so why are some regions blue and others red? I would use a monochrome color map.
- More details explaining figure 2 would be helpful. E.g. it my understanding that the middle row plots the energy of $\rho (\cos \phi v_1 + \sin \phi v_2)$, where $v_1 = (0,1,0,1)$ and $v_2 = (0,1,0,-1)$ are the flexible modes of the linkage.
- Something is wrong with the bottom row of SI figure 2. For example, the energy plot in figure 2B would suggest that the energy of the inner circle at $\phi=0$ is nearly identical to that of the outer circle at $\phi=\pi/2$, which is patently untrue from inspecting the middle disk plot. Similarly the middle plot of figure 2C contradicts the bottom plot's indication that the minimum value of the outer ring is greater than the maximum value at the inner ring.
- Where does the scaling $\rho_c \sim \sqrt{\tilde{\kappa}/K}$ come from? What justifies this law?
- Equation three should be a vector equation, no? There are three constraints? Is E_{Vertex} the sum of the listed energy over the three constraints?
- $C_{ai} \rho^i$: the subscript/superscript convention here conflicts with elsewhere in the text.
- I don't believe the axes in SI figure 3 Surely the energy minima should occur at $\phi = 0$ and $\pi/2$, in the left-bottom plot?
- In equation 7, how are the "faces" terms chosen? Is a spanning tree built on the edge graph of the origami pattern?
- I don't understand the LHS of equation 8: τ_{relax} cannot be a "folding relaxation timescale" because τ must have units of momentum?
- In equation 10, why is there a factor of 1/2? The actuation model assumes that $a(t) \rightarrow 2$ as $t \rightarrow t_{\text{target}}$?
- In SI figure 5, why are the random crease distortion samples clustered so tightly together?
- "an efficient polynomial algorithm": yes, though the most popular numerical code (the simplex method) runs in worst-case exponential time.
- I don't understand what is means for ρ_D to be an "angular minimum" (page 14). I also don't understand the notation in the denominator of equation 16.

Reviewer 1:

This is an extremely interesting paper about how to design origami (or more generally linkages of bars, but the focus is on origami) to select a particular folding pathway so it reaches the desired target state under a wide range of forces. Some recent work by different groups including the authors' has established that starting a from a flat state, there are usually a great many different pathways that a creased paper (with no energy on its creases) can follow, and since all but one of these is the pathway one wants to follow, it should take a highly controlled, precise force to be able to follow this pathway. This is true even if one designs the origami to have a one-degree-of-freedom folding pathway, as the flat state is a highly singular state where many different one-degree-of-freedom pathways intersect, including the designed pathway and lots of undesired pathways. So the goal of designing self-folding origami (i.e. origami which folds spontaneously under thermal fluctuations, or else under a wide range of external forcings) might seem hopeless. However this paper offers an intriguing and in hindsight quite simple idea: make the crease stiff, and with different stiffnesses, and then the energy will change by different amounts along different pathways and so presumably only one pathway will be the lowest in energy initially, and this is the pathway that will be followed when external forcing is applied.

This idea seems so simple, and yet the authors argue quite convincingly that it is effective: they show that one can solve for the required stiffnesses by solving a linear programming problem, hence efficiently, and they demonstrate through simulations under several different kinds of external forcings that this strategy frequently allows the system to follow the desired pathway and end up at the desired folded state. In addition they argue that the number of creases that must be made stiff, is usually much smaller than the number that must be activated if the only control one has is in applying forcings to creases.

I think this is an excellent paper, with an original idea proposed to solve an important problem not only because self-folding origami is a promising way to manufacture materials, both on the macroscale and possibly also the nanoscale but also because the problem of designing assembly pathways occurs in a wide range of materials and statistical mechanics problems, where a system does not always have time to find its ground state so the pathways it follows can determine what it assembles into while the problems take different forms in different contexts, there may still be a link between the strategies used in design. The work is competently and thoroughly carried out, and speculates on the wider implications of the work, making links for example from their observation that different rates of strain choose different pathways, to non-Newtonian materials which have strain-dependent rheologies, an idea I found intriguing. I would recommend publication after some small changes.

We thank the reviewer for the very kind words recognizing the novelty and potential impact of our work. We hope that the method presented in this paper could help alleviate the flat state issues in self-folding origami, providing for practical implementations of multi-functional patterns and the ability to easily fold nanoscale materials manufactured at the flat state.

I do have some small questions/concerns listed below but they are for the most part relatively minor.

Please proof-read while the paper is overall well-written there are a few blatant typos.

We thank the reviewer for the careful reading of our manuscript. We have made efforts to correct the typos in our revised manuscript, and will keep looking out for them and for other ways to improve the language of the paper.

How does one know there is a solution to the linear programming problem setting the creases (Eq 6)? And if there is a solution, how does one know that there is some tolerance around this solution, so that if one doesn't manufacture exactly the stiffnesses one predicts are optimal, the structure can still fold along the desired pathway? Or in other words, how does one know that the volume of possible solutions in kappa-space is large enough? If the number of undesired pathways increases exponentially with the number of vertices, then isn't it possible that the volume of solutions for a single pathway in kappa-space decreases in a similarly exponential fashion?

This concern would be important to address to make sure the strategy the authors suggest is actually feasible in practice.

There may be a simple argument to argue that there is actually quite a lot of tolerance on the creases for example the number of edges (E) is about twice the number of vertices (V), so if there are roughly 2^V pathways and roughly 2^E possible solutions in kappa-space (if to a rough approximation the linear programming solution selects a subset of creases to activate), then there are more than enough choices of edges to cover the pathways with some tolerance. However, I'd be interested in what the authors think.

The reviewer raises an important question about our method, regarding solutions obtained by Linear and Quadratic Programming. The reviewer's argument is indeed the first step; there are 2^E qualitatively different solutions in kappa space (i.e., each crease is either stiff or not stiff) while the number of branches grows as 2^V . However, two different pathways could possibly share the same qualitative kappa solution but differ in the precise stiffnesses needed. Hence, as the reviewer points out, we must consider the robustness of such solutions to manufacturing errors in the stiffness profile κ .

We simulated such errors by perturbing the optimal solutions by relative values of up to 5% (for small and large patterns). We find that the resulting unique low- ρ mode is essentially a perturbed version of the one obtained for the optimal solution; in particular, the critical saddle-node folding magnitude ρ_c is changed. Undesired modes are still lifted near the flat state.

We have added a paragraph in the end of SI section IV to discuss such tolerance issues.

How does one know what the transition state is (eg in Eq 2)? And for a material with lots of creases, there should be many transition states they all appear in the minimization problem (Eq 6), so presumably must all be found how does one find them? This sounds like a hard problem.

We thank the reviewer for pointing out this subtle but crucial part of our argument. Indeed, finding transition states (or even minima) in a complex high-dimensional landscape is

a hard problem. We make use of a crucial simplification briefly mentioned in the manuscript: If the desired mode is a low energy self-folding mode that nearly solves loop equations, it is approximately consistent with an independent binary choice of mode for each vertex. Therefore, one could approximately locate both the high dimensional transition states in terms of combinations of the transition states of single vertices, as seen in Eqn 6, our generalized saddle-node constraint for larger patterns.

I didnt understand how Eq 2 ensures that the energy of $\tilde{\rho}_D$, is lower than both the transition state and the undesired mode. Eq 2 seems to be an equation which says that the energy of the undesired mode, is greater than the energy of the transition state. But says nothing about the desired mode. Maybe I missed something? Perhaps the authors could clarify, either in their response, or the paper?

We thank the reviewer for raising this point; we have now clarified it. Indeed, Eq 2 only requires that all undesired modes are eliminated in bifurcations with the transition state. The desired mode might also be eliminated, especially if random stiffness profiles that satisfy Eq 2 are used. We have updated the text below Eq 2 to say:

”Any set κ_i satisfying the above constraints that predominantly raises the undesired mode will eliminate it in a saddle-node bifurcation.”

Instead, it is our minimization procedure (and not Eq. 2) that is aimed at protecting the desired mode from being eliminated. For example, the LP method minimizes the energy of the desired mode, biasing us towards solutions κ that lift and eliminate the undesired modes without eliminating the desired mode. This is not a mathematical guarantee; the few failed cases discussed in the SI are of this type. However, it appears to work for most patterns as shown in Fig 3.

In Fig 2C there is a red point which appears to have lower Distortion than any others (in the south-west corner). How is this possible, if the filled-in red dot was found by optimizing over distortion?

We believe the point referenced by the reviewer (in the south-west corner of Figure 2c) is part of the figure legend, and not an actual data point. To avoid such confusion, we wrapped the legend with a box.

- SI Fig 2 can you say something about the color scheme?

The reviewer is quite right in pointing out that SI Figure 2 was poorly presented, a shortcoming that brought about several misunderstandings. The revised version of the SI has updated both the figure itself and how it is presented in the caption and the surrounding text. The colored contours represent the scaled null space energy in a way meant to emphasize the low energy paths in configuration space as blue valley surrounded by red ridges. We believe the current manuscript describes this idea more clearly, as well as the precise definitions of ρ and ϕ in the context of the 4-bar linkage.

We hope our edits and re-done figures address all the points raised by the reviewer. We thank the reviewer for the kind words and for helping improve our manuscript.

Reviewer 2:

The paper under review considers a pitfall of self-folding mechanisms, framed in the context of mechanical metamaterials, whereby such a mechanism could fold into an undesirable state. That is, the configuration space of such a mechanism may have branch points, and in order to control the mechanical behavior, one wishes to guarantee movement along the desired path through such branch points. In this paper, we see a new method for such mechanical control via the utilization of intentional imperfections in the folding joints, e.g., making some joints more stiff than others. The authors describe how such imperfections can be used to tune the mechanics and thus avoid undesirable folded states.

This work is quite excellent. The authors' analysis of varying stiffness of folding joints is complete and, to my eyes, correct. They consider the 1-dimensional linkage case as well as 2-dimensional folded sheets. While they restrict themselves to 4-bar linkages in the 1D case and degree-4 vertices in the 2D case, these cases are the most immediately useful to current engineering applications, and since such systems are 1-DOF, they make the most sense to consider initially. Aside from some comments on the paper's exposition (see below), I see no flaws in this work.

This paper is certainly of great interest to researchers in the field of folding mechanics. While it does serve as an interesting lesson of what can happen in metamaterial design, it really is specifically about folding- or origami-inspired metamaterials. Any researchers who have been following the surge of interest in folding in the physics and engineering communities will find this paper of significant value. It will definitely make a positive change/impact on the field.

We thank the reviewer for the encouraging words, and pointing out the relevance of the work to the burgeoning field of self-folding meta-materials. We hope that this work will inspire engineered applications that overcome the flat state exponential branch point and produce truly self-folding structures.

After some thought, I am deciding to recommend this paper for publication in Nature Communications. However, I want to explain my hesitation, because I think it suggests a slight change in emphasis by the authors.

When I first read this paper, I was intrigued but kept thinking to myself, "This is not surprising at all." The idea that adding stiffness to creases would alter the configuration space, especially if one incorporates the total strain into the configuration space, is completely unsurprising. Thus, while the work in this paper is very complete, solid, and technically new, I initially thought that I might not recommend it for publication because the over-all conclusion seemed somewhat obvious.

But further examination of the results changed my mind for the following reason: The addition of stiffness to the folding joints can be used to tune the configuration space even if different folding torques are applied to the joints. This result is, I think, not emphasized in the paper; it is only mentioned in the short "Other models of folding forces" section on pages 5-6. I will comment on this in some more detail below, but the point I want to make is that this aspect of the authors' stiffness approach is, to me, quite surprising and useful. That is, one could argue that adding joint stiffness is

tantamount to using different torques at the joints (which has been studied previously), that these two different approaches might be, at least mathematically, equivalent. E.g., one could model each with a vector field on the configuration space of the folding angles ρ_i . However, the results of the simulations shown in the SI Figure 4 show a very high success rate of desired self-folding over a variety of joint torques with stiffness added versus low success rate without the added stiffness.

Therefore I strongly suggest that the authors call more attention to this aspect of their results. I think that engineers in particular will find it interesting that adding joint stiffness seems to give a much better rate of successful self-folding than programming specific torques for the joint actuators.

Therefore I do recommend this paper for publication in Nature Communications.

We thank the reviewer for raising this important point about presentation. We agree that this feature is one of the most interesting and also experimentally relevant aspects of our result - that the modification of configuration space is robust to any folding forces applied as well and hence success with folding forces is dramatically improved by crease stiffness. As the reviewer notes, material supporting this point was obscured in the Supplementary Information. We have taken multiple actions to highlight how our approach changes the configuration space in the context of folding forces as well:

1. New figure 4: We have created a new figure 4 using the earlier supplementary figure referred to be the reviewer. The figure and caption are entirely dedicated to making the point described above.
2. New expanded section accompanying new figure 4: We have a section about this figure, explaining how the modified configuration space is robust to addition of forces and thus dramatically promotes successful folding when external folding forces are applied.
3. Expanded discussion of relationship to work by Tachi and Hull: In the new section accompanying figure 4, we also explain the similarities and differences to the approach of Tachi and Hull.
4. We also emphasize the point raised by the reviewer in the introduction: *Designing the topology of the bifurcation diagram presents several benefits. Once this topology has been designed for a material, it is not modified by entire classes of applied folding forces but determines the response to such forces.[...]*

and briefly in the abstract: *Our approach allows robust folding by entire classes of external folding forces.*

and in Sec I.B: *To solve the misfolding problem for diverse folding methods, our approach intentionally ignores specific external folding forces when reprogramming the topological connectivity of modes. Since folding success relies on the topology of a bifurcation diagram, our results are mathematically robust to several classes of folding forces as shown later.*

Below are more specific comments and suggestions for the authors.

(1) The introductory paragraphs of the paper do not do a good job of preparing the reader for the main topic of the paper. That is, nowhere is it specifically stated in the first 4-5 paragraphs that the authors will be studying folding processes. In fact, the first indication that the subject of the paper is folding mechanisms is at the end of the 4th paragraph when suddenly "joint stiffness" is mentioned — nowhere before is it stated that the materials being considered are folding along joints or crease lines. Even the title of the paper does not suggest this. (The abstract does, but the main body of the text should be written to be understood independently of the abstract.) I understand the desire to highlight the general connection to mechanical metamaterials, but the paper should state up-front that the main subject is folding.

We have now rewritten our introduction to address this issue. We now introduce the specific systems - creased sheets and elastic networks - in the first paragraph. We then describe the specific problem of folding self-folding origami and its branch point in the second paragraph: *[..] self-folding origami, despite the name, has an exponential number of misfolding pathways that meet at a 'branch point' at the flat state [], making it nearly impossible to fold into the desired folding mode.*

We also explicitly state the systems of interest before talking about joint stiffness: *We focus on elastic networks and creased sheets where rods or plates are connected at flexible joints. We show that heterogeneous stiffness in such joints can completely change the [...]*

We rewrote the title to explicitly invoke 'folding' : *'Shaping the topology of dynamical folding pathways in mechanical systems'*

We agree that stating the concrete problem covered the paper early on is important for clarity. We thank the reviewer for helping improve our presentation.

(2) A minor gripe about notation: The variable ρ is used for the total strain of the system, and at the same time other variants of ρ are used for other variables. E.g., in the SI, $\vec{\rho}$ is the vector of coordinates for points of a 4-bar linkage, and ρ_i is the folding angle of crease i in a 2D folded crease pattern. This is a bit confusing. Maybe a different variable could be used for the total strain?

As the reviewer points out, the notation of SI section 1 regarding the ρ variable was confusing. The current version reworked the notation for that section, with the 4-bar configuration described by a new variable P , and the strain variable ρ is defined consistently with the origami case.

(3) I remain confused as to what the variable ϕ represents. The SI describes it (in the 4-bar linkage and degree-4 vertex cases) as the "mixing angle of the two zero-modes spanning the 2-dimensional null space." What is meant by "mixing angle"?

The reviewer is justifiably confused. Our old manuscript did not define ϕ explicitly, instead just naming it a 'mixing angle' of two modes. We have now explicitly defined this angle twice in the SI, once in each context it appears (for the 4-bar linkage and the origami vertex). We hope that this approach will satisfy both readers who want to study the details of the energy model, and the readers who are more interested in the bifurcation theoretic description (for which the exact parametrization of the energy is less important).

(4) The first page of the SI seems to have some typos. In the sentences after equation (1) we see: $\vec{\rho}_0 = [x_B = L_1, y_B = 0, x_C = L_1 + L_2, y_B = 0]$ Shouldn't the last term there be " $y_C = 0$ "? And later in the same paragraph the authors write, "To see this, note that to lowest order in ρ , the two zero-energy motions are..." Should that be $\vec{\rho}$? From the main text of the paper, I am still assuming ρ is the total strain, yes? That doesn't seem to be what you mean here, or is it? In the next paragraph is mentioned ρ^4 . Is this also meant to be the total strain? Or should it be some part of $\vec{\rho}$? Or maybe $\|\rho\|^4$? This notation is also present elsewhere, such as in the caption to SI Figure 1 and 2.

We thank the reviewer for identifying and communicating the issues of notation (and typos) in SI section 1. As mentioned in the response to (2), the current SI revises this part with revamped notation and more details to improve clarity.

(5) Perhaps I am missing something, but I am surprised that the relative lengths of the creases in the 2D case (Section B in the main paper) do not seem to be taken into consideration. Adding stiffness to a longer crease versus a shorter crease should have a different impact to the energy landscape, yes? Perhaps the relative lengths of the creases are captured in the energy model, but if so, I don't see how this is done. Perhaps a clarifying sentence addressing this is needed somewhere.

The reviewer raises an important point regarding any practical implementation of the method described in this work. We treat crease stiffness as a simple modulus that multiplies the crease folding angles (squared). Realistic models of thin creases made of some material will resist folding with a modulus that scales linearly with crease length. Thus, when designing the stiffness modulus of a crease, one would have to account for its length.

Suppose the crease is of length l and thickness t . It is known that the bending modulus of the material scales as its thickness cubed t^3 (Witten, Rev. Mod. Phys 79, 643 (2007)). Thus, the stiffness of the crease will be $\kappa \sim lt^3$. One can set the stiffness modulus of the crease of length l by properly modifying its thickness during the manufacturing process.

To clarify this point we now say in the SI, near the end of the vertex section, "*In practical applications, the folding stiffness moduli of the creases κ_i will depend on each crease's thickness, length and material properties. Generally the length and material are given by the type of application, so that the value of κ can be set by proper choice of the crease thickness t . It is known that for general elastic materials $\kappa \sim t^3$.*"

(6) In the SI, Section III.A., when describing their simulations of torque-based folding, the authors state, "It is hard to call such high dot-based folding 'self-folding' since such actuation requires a large number of actuator creases and the torques F_i on each crease needs to be tuned carefully." However, the authors seem not to be considering the approach in the paper:

Tomohiro Tachi and Thomas C Hull, Self-Foldability of rigid origami, Journal of Mechanisms and Robotics, 9 (2), 2017, 021008-021017.

(which note is not the same paper as the authors' reference [35]). In that paper an approach to actuator-based self-folding is made whereby the torques are tuned to make low (or, if possible, zero) dot product with the undesirable configuration space branches.

This greatly reduces the chance that the undesirable branches will be followed. The authors' point may remain the same, but perhaps programming torques to have low dot product with the tangents of undesirable branches would have a better success rate in the absence of added joint stiffness.

As noted earlier, we have now added an entire section discussing such torques and how they are distinct from our stiffness based approach. Indeed, there is a mathematical similarity in that both approaches add a quadratic function to the underlying landscape. However, our results are distinct because our quadratic function is centered at the singularity while Tachi and Hull have placed it away from the origin; at the branch point, their approach effectively adds a linear potential which is not equivalent to our approach. In particular, Tachi and Hull require undesired and desired branches to have non-positive dot products (unlikely to be the case for branches of larger patterns). However, our method works even for those cases. We thank the reviewer for asking us to clarify the similarities and differences between these approaches.

(7) In the references of the main paper:

* In reference [2], the journal Science needs to be capitalized. * reference [20] seems to be a duplicate.

We thank the reviewer for pointing out these issues with the references. They were fixed in the revised manuscript.

Reviewer 3:

Rigid-foldable origami patterns made of quadrilateral faces and valence-four vertices generically have one degree of freedom: actuating one crease edge folds the entire pattern. The same is not true when the pattern is in the flat configuration: multiple zero-energy folding modes exist in the flat state, so that in practice origami must be pre-creased with mountain and valley folds to bias the pattern towards the desired folding mode. This paper studies ways of eliminating undesired folding modes by adding torsion springs to a sparse set of crease edges; if chosen carefully, these springs modify the energy landscape in a small neighborhood of the flat state so that there is a unique elastic-energy-minimizing mode, guaranteeing that the pattern, when quasistatically folded, will fold in that mode. The paper presents two algorithms for selecting springs to add: one minimizes the energy (which, in the ideal case, is zero) of the desired folding mode, while prohibiting all others; the second attempts to minimize the geometric distance between the desired folding mode of the unmodified origami pattern, and the perturbed configuration that would actually be seen when folding the pattern with added springs (these springs introduce additional internal forces that will be balanced by strain at the faces and edges of the origami pattern during folding). The former leads to a linear programming problem, and the latter, a quadratic programming problem, both of which can be easily solved using numerical techniques. The paper demonstrates that the optimized springs achieve the goal of minimizing energy and geometry distortion far better than randomly placing springs.

Overall, I agree that the problem of taming the complex and degenerate energy landscape of origami patterns is an interesting and important problem whose solution would greatly improve the practical usefulness of folding structures to engineers and materials scientists, by making the structures more controllable. The approach suggested by the paper, though simple, seems reasonable and was validated on some large origami patterns.

We thank the reviewer for the detailed comments. We are happy the reviewer finds our work interesting and contributing to a fundamental question in the field.

That said, I do have some technical questions and concerns:

1) Equation 2 is the heart of the paper, as it is used in both the LP and QP optimization algorithms to try to ensure that for small strains, only one folding mode away from the flat state is in static equilibrium. But equation 2 doesn't quite achieve this goal: adding springs does increase the energy of the initially-zero-energy undesirable configuration $\tilde{\rho}_U$, but also shifts the energy minimizer $\tilde{\rho}_U^*$ away from $\tilde{\rho}_U$: the energy of $\tilde{\rho}_U^*$ could be less than E_{TS} even though $\tilde{\rho}_U$ has greater energy than E_{TS} . See for instance SI figure 3B and 3C, where it is evident that the location of the energy minima on the ϕ -axis changes as ρ increases. Perhaps a "safety factor" should be added to the right-hand side of equation 2; it would be even more satisfying if this shift in minimizer were accounted for theoretically.

We thank the reviewer for bringing up this subtle point regarding Eq. 2. It is true that the addition of stiffness shifts the minimizer $\tilde{\rho}_U^*$ away from $\tilde{\rho}_U$, and quite strongly in the region of the saddle-node bifurcation. Unfortunately, due to the complex nature of the origami

potential, this shift is hard to account for theoretically. We instead employ a "safety factor" as suggested by the reviewer. This is done by understanding that ρ_c , the folding magnitude for which the saddle-node bifurcation appears, is just a scale and not an exact number. To set ρ_c one would need to find the precise "safety factor", which would depend on the geometrical details of the pattern. No matter what choices we make, the saddle node bifurcation will happen at some $\rho \sim \rho_c$. For the purpose of removing the bifurcation up to a distance $\sim \rho_c$ from the flat state, Eq. 2 is sufficient.

2) Geometric distortion is quantified (in the QP optimization) by looking at the energy gradient at the desired (and, initially, zero-gradient) folded configuration. Why is the gradient a good surrogate for geometric distortion? This choice seems dubious to me: it essentially assumes that the Hessian of energy does not change much near ρ_D (reasonable) and that the Hessian is isotropic there (seems unlikely). Were any experiments performed to measure how closely the gradient magnitude correlates to true distance (in the Hausdorff sense, say) between the unaugmented and spring-augmented desired folding modes of the pattern? Are there any theoretical reasons to believe that the gradient is a good measure of geometric distortion?

Geometric distortion is defined by the dot product between the desired mode and the actual minimum obtained due to stiff creases near the flat state. As the reviewer points out, to optimize geometric distortion we make use of the angular part of the gradient at $\rho = \rho_D$, and choose the stiffness profile to make it as small as possible (subject to the constraints). The reviewer is right in pointing out that the gradient is just a surrogate for distortion, and perhaps not the most obvious choice. While it is understood that any heterogeneity in the stiffness profile κ_i results in geometric distortion of the mode, it is not clear whether optimizing the gradient (subject to constraints) actually results in the least distorted mode.

Our particular function is quadratic in κ_i , so it can be optimized efficiently using Quadratic Programming algorithms. Numerical evidence (presented in the paper) establishes it as a useful heuristic that gives rise to the least geometrically distorted modes, by a large margin, compared to any other prescriptions we tried.

While more sophisticated formulas might show lower distortion theoretically, we focused on a simple function that also has an efficient algorithm for minimization.

In terms of exposition, the paper is mostly clear, albeit only when interleaving reading the main paper with the SI. I wish there had been more results about folding-rate-dependent structures (I.C), as this is a very intriguing yet underexplored mechanism for controlling deployable structures, and less space spent on the four-linkage example, which it is not substantially simpler to understand than the four-crease origami unit cell of section I.B. Some detailed comments:

- the paper correctly points out that the proposed method relies on bifurcations existing only at the flat state. For rigid-foldable flat origami this assumption seems reasonable, but it would be useful to have more discussion about when, in practice, this assumption is likely to be invalid.

As the reviewer correctly points out, in general bifurcations do not have to all happen at a single point in the configuration space. A new branch point could happen somewhere

else, e.g. in more complex linkages. To deal with multiple such points, one would need to repeat our analysis for stiffness about each of them. It is not clear how the stiffness profiles needed for the multiple branch points would affect each other; though we suspect that the stiffness used for each branch point looks like a linear potential for the rest. We argue such linear modifications do not change the topology much. Investigating multiple branch points would be very interesting but beyond the scope of the current paper.

- More explanation of figures 1B and 1D would be useful. What is ϕ_{TS} ? Why do the left panels of 1B and 1D use a different vertical axis than the right panels (energy cross-sections)? Why are some dots in 1C bigger than others (this is explained in the SI but not, I believe, in the main text or figure caption.)

- It is stated that $\rho = \|\vec{\rho}\|$ in the top-right of page 2, but ϕ is used heavily beforehand (including in Figure 1 and the SI) and it would have been useful to have this definition earlier.

We thank the reviewer for communicating that we failed to define ϕ clearly in the original manuscript. We have now explicitly defined this angle twice in the SI (once in each context it appears, for the 4-bar linkage and the origami vertex). We hope that this approach will satisfy both readers who want to study the details of the energy model, and the readers who are more interested in the bifurcation theoretic description (for which the exact parametrization of the energy is less important).

The use of different vertical axes is chosen to illustrate the main idea of the paper to readers from different field, mechanics and bifurcation analysis: the right panels clearly show the existence of two different minima in the energy landscape of the free-folding linkage, or one minimum for the stiff linkage. The middle panel used a different parametrization of the fixed points of the right panels in order to show the bifurcation structure in a familiar way to readers from that field. Adding stiff hinges is shown to change the bifurcation diagram structure from a pitchfork bifurcation to a single minimum and a saddle-node bifurcation.

As for the dot size in Figure 1c, we now say in the caption *"larger orange circles denote stiffer springs."*

- "our approach intentionally ignores specific folding forces": that is to say, the forces driving folding, not the elastic forces arising from folding.

Indeed. The strength of our method is that it designs the bifurcation structure of folding space such that most reasonable folding schemes (i.e. external forces) will find the desired mode. The internal forces of the sheet are due to its geometry (vertex and face energy terms) and our chosen crease stiffness profile.

To make this point clearer, we now say on Pg 3 *"our approach intentionally ignores specific external folding forces when reprogramming the topological connectivity of modes."*

- the main text calls an elastic formulation that models rod stretching the "alternative model," but in the SI it is the bending model that is "alternate."

- speaking of the bending model, it is not spelled out in detail (presumably the rods are now assumed exactly inextensible, but can deflect (with what profile?) between the endpoints?) and it is not clear to me that the analysis is identical to the case of Hookean unbending rods.

We have fixed the ambiguity of what is the 'alternative model' in the text. Now we only refer to the bending model as alternate. We have looked at a bending model, and though it was not analyzed to the same extent as the simpler compression model, it numerically leads to the same phenomenology (i.e. two zero-modes spanning a linearized null space whose energy scales as ρ^4).

Since the flat state bifurcation in this system arises due to the structure of the constraints, we believe that any reasonable energy model used to replace these constraints will result in null spaces with similar structures.

- The two bending modes in 2A appear to be the same mode (with different sign); this may be an artifact of the rendering but it would be useful if these two modes were more clearly different.

We have now numbered the creases of the vertices in Fig 2a. With this updated figure, we believe it is clear that the creases are folding in completely distinct ways.

- In figure 2C, how is geometric distortion measured? Page 5 mentions having "low dot product" but that does not clarify for me.

The geometric distortion is measured by taking the normalized dot product between the desired mode ρ_D and the actual unique minimum close to the flat state ρ_D^* . We define geometric distortion as $GD = 1 - \tilde{\rho}_D \cdot \tilde{\rho}_D^*$.

To clarify this, we now say on the main text "*Geometric distortion*", defined by one minus the normalized dot product of the desired mode and the obtained minimum."

- I don't understand the frontier of black dots and lines in figure 2C. What is ρ^{-n} ?

We thank the reviewer for pointing out this presentation issue.

To clarify, we now say in the main text

"The black line $\kappa_i \sim (\rho_D)_i^{-n}$ for positive n , shows that stiffness profiles generally trade-off energetic and geometric distortion."

In the SI section IV we now say:

"LP and QP stiffness protocols achieve optimized distortion values better by orders of magnitude compared to other schemes, including the ρ_D^{-n} prescription that worked well for single vertices."

- A sentence or so explaining "the vertex constraint" (page 3) in the main text would be useful.

We now explain,

" E_{Vertex} accounts for bending of vertex faces..."

Additional explanation about this energy term is in SI section II.

- What does it mean for $\tilde{\rho}_U$, etc to be "normalized"?

To make this point clear, we now clearly state that "normalized" means of unit norm in the main text.

- The notation in equation 4 is rather sloppy; presumably bold-face ρ , squared, signifies component-wise squaring of all entries of $\vec{\rho}$. I'd prefer notation like \odot or \star for component-wise multiplication, with a few words of explanation.

The reviewer is correct in commenting about the presentation of Eq. 4. We have reformulated it in the revised manuscript to be mathematically rigorous and similar to other equations (Eqs. 2,6).

- "the optimal stiffness profile is guaranteed to be sparse": why is this?

In Linear Programming, the optimization function (Eq. 4) is linear in κ_i and composed of equivalent planes. These linear constraints (Eqs. 2-3) define a simplex, $\sum_i a_i \kappa_i \geq b, \kappa_i \geq 0$, whose boundary is in the coordinate planes of the κ space. The solution to convex Linear Programming problems are known to occur at a boundary vertex of such a simplex, which will generically have several $\kappa_i = 0$.

To clarify this point, we have cited the relevant references for Linear Programming on Pg. 4.

- What is a "radial component" of the gradient (equation 5)?

We thank the reviewer for pointing out this vague reference in the original manuscript. The revised manuscript now reads "*after projecting out the component of the gradient in the ρ_D direction.*"

- Section 1 in the SI interchangeable treats the linkage as having one degree of freedom, with three rigid kinematic constraints; having four degrees of freedom, with two points A and D pinned; and having the full eight degrees of freedom (in equation 2, eg). It would be less confusing to make the parameterization of the problem more consistent and explicit.

We thank the reviewer for the detailed reading and analysis of our Supplementary Information. In the beginning of the SI section we treat this system as a 1 degree-of-freedom system, with 4 variables and 3 constraints. However, we then wish to relax these constraints by replacing them with high energy costs due to a large compression modulus K . This idea is explicitly articulated in the SI, and leads to "recovery" of the bar length degrees-of-freedom.

To avoid this ambiguity, we now say "*When constraints are allowed to be broken by compression (or bending) of the rods, allowing nonzero energy configurations in the full 4-dimensional $\vec{\rho}$ -space...*"

- Why is energy quartic in ρ^4 , close to the flat state? I agree with the surrounding sentences (that energy due to extension vanishes to quadratic order, etc) but I do not see how it follows that the cubic term vanishes.

Our origami model is symmetric to reflection about the flat state plane (switching signs of all ρ_i). Energy terms cubic in ρ_i are inconsistent with this symmetry.

- More details explaining figure 2 would be helpful. E.g. it my understanding that the middle row plots the energy of $\rho(\cos \phi v_1 + \sin \phi v_2)$, where $v_1 = (0, 1, 0, 1)$ and $v_2 = (0, 1, 0, -1)$ are the flexible modes of the linkage.

- Something is wrong with the bottom row of SI figure 2. For example, the energy plot in figure 2B would suggest that the energy of the inner circle at $\phi = 0$ is nearly identical to that of the outer circle at $\phi = \pi/2$, which is patently untrue from inspecting the middle disk plot. Similarly the middle plot of figure 2C contradicts the bottom plot's indication that the minimum value of the outer ring is greater than the maximum value at the inner ring.

- The color scheme in SI figure 2 is confusing (especially in black and white): energy is everywhere-positive so why are some regions blue and others red? I would use a monochrome color map.

We have now addressed these issues with SI Fig 2. The revised SI version has more detail and revamped notation that better define and explain the variables ρ and ϕ and the vectors spanning the null space.

Color scheme: The revised version of the SI has updated both the figure itself and its description in the caption and surrounding text. The colored contours represent the *scaled* null space energy in a way meant to emphasize the low energy paths in configuration space as blue valley surrounded by red ridges.

The apparent inconsistency pointed out by the reviewer was caused by our failure to communicate what contours were plotted in the previous version of the manuscript.

We believe the current manuscript describes these ideas more clearly, as well as the precise definitions of ρ and ϕ in the context of the 4-bar linkage. We hope that using colored contours instead of the continuous color map makes it easier to see the topology of the low energy modes in configuration space.

- Where does the scaling $\rho_c \sim \sqrt{\tilde{\kappa}/K}$ come from? What justifies this law?

We have now expanded this part to argue that the critical value ρ_c arises where the energy due to stiff hinges (or creases) is comparable to the intrinsic energy of the system. As the stiffness energy scales like $\tilde{\kappa}\rho^2$ and the intrinsic energy scales like $K\rho^4$, they are comparable when $\rho_c \equiv \rho \sim \sqrt{\tilde{\kappa}/K}$.

- Equation three should be a vector equation, no? There are three constraints? Is E_{Vertex} the sum of the listed energy over the three constraints?

We have made our notation clearer to show this explicitly on Pg. 5. The reviewer is correct in pointing out that E_{Vertex} is the sum of energies due to breaking each one of the 3 independent vertex constraints.

Eq. S4 referred to by the reviewer is not cast in a form of a vector equation, but as a matrix equation. However, it turns out that of the 9 equations, only 3 (corresponding to the upper or lower triangle of the matrix) are nontrivial and independent. This means the constraint equations can be cast in vector form $T_a = 0$. We have chosen to skip this step as we feel it is not sufficiently informative.

- $C_{ai}\rho^i$: the subscript/superscript convention here conflicts with elsewhere in the text.

We have now fixed this notation to be consistent.

- I don't believe the axes in SI figure 3. Surely the energy minima should occur at $\phi = 0$ and $\pi/2$, in the left-bottom plot?

We have revised the manuscript to clarify this presentation issue. The definition of ϕ used in the caption and that used in the figure are different by a constant offset.

Some extra details: For origami vertices, we find the linearized null space by numerically performing Singular Value Decomposition on the constraint matrix of Eq. S4 (at the flat state). By this process we obtain two perpendicular eigenvectors that span the linearized null space in which the two zero-modes live. However, the numerically obtained eigenvectors do not have to coincide with the zero-mode directions. In that case, the special zero-mode directions occur at some linear combination of these eigenvectors (implying a nonzero mixing angle ϕ). As the reviewer correctly notes, the angular difference between the two zero-modes in this space is indeed $\pi/2$, even if the modes are not located at $\phi = 0, \pi/2$.

- In equation 7, how are the "faces" terms chosen? Is a spanning tree built on the edge graph of the origami pattern?

Face bending terms due to the loop constraints are simulated by augmenting the origami pattern with diagonal creases, one for each inner face. These diagonal creases are associated with springs of stiffness due to the face bending modulus κ_f . The pattern is then folded with those stiff creases, such that the face energy term are given by the amount of folding of the diagonal creases. This model was explored in depth in *Stern et al., PRX 7, 041070 (2017)*.

We have now added a citation to this effect.

- I don't understand the LHS of equation 8: τ_{relax} cannot be a "folding relaxation timescale" because τ must have units of momentum?

Eq S8, and all other equations regarding the folding dynamics, are written in units in which the stiffness moduli κ are dimensionless. In this system of units the speed of folding $\dot{\rho}$ is proportional to the sum of forces applied on the current configuration (overdamped dynamics). The relaxation timescale τ_{relax} is then required to fix the units, in a similar way to the use of a friction or resistance constants in other damped systems.

Furthermore, irrespective of choice of the system of units, overdamped dynamics imply the existence of a typical timescale in the system due to its damping relaxation time.

- In equation 10, why is there a factor of 1/2? The actuation model assumes that $a(t) \rightarrow 2$ as $t \rightarrow t_{target}$?

We thank the reviewer for pointing out this mistake in the equation. The 1/2 factor was supposed to be outside of the sum to cancel the 2 factor due to the gradient.

In the revised manuscript, the force due to the corrected potential is that of springs whose targeted rest angles change with time.

- In SI figure 5, why are the random crease distortion samples clustered so tightly together?

The reviewer raises an interesting point. In figure S5a we have sampled a relatively small number of random stiffness profiles satisfying the constraints of a quad pattern (made of a loop of four vertices). We observe that even though all but one mode are lifted close to the flat state, the unique one remaining is highly distorted (both in energy and geometry).

This effect seems to be an example of the "curse of dimensionality". While for single vertices, the space of stiffness profiles is only 4-dimensional, this space rapidly grows in dimension for larger patterns. For the quad patterns of figure S5a, the space is 12-dimensional. We find that even though the LP and QP solutions are well optimized and fairly robust to changes in κ , these regions only account for a minuscule, hard to sample volume of κ -space. The vast majority of κ -space gives rise to profiles that considerably distort both energy and geometry of the mode, making them appear clustered in the figure (compared to the vastly better LP and QP solutions).

- "an efficient polynomial algorithm": yes, though the most popular numerical code (the simplex method) runs in worst-case exponential time.

We have now clarified this point in the main text by saying on Pg. 4 that "*Linear Programming problems are solved in polynomial time, as long as an efficient algorithm is used.*"

- I don't understand what it means for ρ_D to be an "angular minimum" (page 14).
I also don't understand the notation in the denominator of equation 16.

Both the term "angular minimum" and ingredients of Eq. S16 are now spelled out explicitly.

We say that the angular minimum is an "*energy minimum at given ρ* ". As for Eq. S16, we now say "*the numerator multiplies the vectors ρ_D and the energy gradient at $\rho = \rho_D$, while the denominator multiplies their norms.*"

- " $L_{AB} \cdot L_{CD}$ ": comma?

- "Equation 2" -> "Equation 2"

We thank the reviewer for pointing out these typos. These were corrected in the revised manuscript.

We hope we have addressed all the comments raised by the reviewer. We thank the reviewer for finding that our work meets the high bar for originality and impact required by Nature Communications.

REVIEWERS' COMMENTS:

Reviewer #1 (Remarks to the Author):

The authors have addressed my comments adequately. I recommend publication.

Reviewer #2 (Remarks to the Author):

In the revised manuscript, the authors have addressed all of my concerns. Therefore I stand by my previous recommendation that the paper be published in Nature Communications.

Reviewer #3 (Remarks to the Author):

I thank the authors for their detailed response to the first round of reviews. Most of my concerns have been addressed in the revision, though I recommend some minor revisions before the paper is published:

1) Reviewer #1 asked about the feasibility of LP and QP problems, and about the sensitivity of the solution under perturbations. The author response addressed the second question, but I still wonder about the first: is the LP/QP always guaranteed to be feasible? Does this require e.g. rigid-foldability of the origami pattern? Were infeasible linear programs ever encountered during the experiments (corresponding to undesirable modes that cannot be eliminated using the paper's stiffening strategy) and if so, how is this situation handled in practice (are some of the constraints removed, until the problem becomes feasible?)

2) Reviewer #1 also inquired about why an analogue to equation 2 is not imposed to ensure that the desired deformation mode has lower energy than the transition state. The reply argues that the objective function's role is to ensure that the desired mode has low energy---while true, this response does not fully answer the question of why inequality constraints were not/could not be imposed. Are such constraints simply redundant? Or are there cases where their inclusion would make the LP infeasible (corresponding to desired modes that cannot be programmed using the paper's proposed strategy?)

3) Figure 1 is clearer in the new revision, though I still have some suggestions about improving its clarity:

- the use of ϕ in this figure (and throughout the main text) is still confusing. Why is $\pi/4$ the transition state, and why are the two configurations shown at the bottom of figure 1a $\phi=0$ and $\phi=\pi/2$? Could a simpler parameterization of the linkage's configuration be used instead (for instance, the interior angle between one pair of rods) for this figure?
- Figure 1c and 1d do not appear to agree: in 1c, the energy minimizer is always at $\phi=0$, whereas from 1d I would expect curve 1 to have minimizer near $\phi=\phi_{TS}$ (presumably slightly less than $\pi/4$), etc.

4) Other minor comments about the text:

- capital K is not defined (page 2, bottom)
- as discussed in previous reviews, it would be useful to include on page 3 a few sentences about how $\tilde{\phi}_{TS}$ and E_{TS} are computed in practice.
- Maybe I'm missing something obvious: why is the RHS of equation 2 E_{TS} , rather than zero?
- the paper mentions that, on the one hand, there is typically a large variation in possible folded states when the strain rate is large; on the other hand, the folded pattern in figure 5c is reproducible, even the fast folded state in figure 5c, right. Is there something special about the example in figure 5c that ensures reproducibility? I would expect fast folding to sometimes yield the "slow" or "intermediate" states instead? More discussion would be valuable.

Reviewer 1:

The authors have addressed my comments adequately. I recommend publication.

Reviewer 2:

In the revised manuscript, the authors have addressed all of my concerns. Therefore I stand by my previous recommendation that the paper be published in Nature Communications.

We thank the reviewers for considering and recommending our revised manuscript for publication in Nature Communications. Their previous questions and comments were instrumental in improving the paper and making it suitable to broad audiences.

Reviewer 3:

I thank the authors for their detailed response to the first round of reviews. Most of my concerns have been addressed in the revision, though I recommend some minor revisions before the paper is published:

1) Reviewer 1 asked about the feasibility of LP and QP problems, and about the sensitivity of the solution under perturbations. The author response addressed the second question, but I still wonder about the first: is the LP/QP always guaranteed to be feasible? Does this require e.g. rigid-foldability of the origami pattern? Were infeasible linear programs ever encountered during the experiments (corresponding to undesirable modes that cannot be eliminated using the paper’s stiffening strategy) and if so, how is this situation handled in practice (are some of the constraints removed, until the problem becomes feasible?)

As we explain in Supplementary Note 4, we generally obtain feasible solutions that saturate the constraints. The mathematical existence of solutions (i.e. feasibility) to the LP and QP problems presented in this work is guaranteed by the simple structure of the constraints. The constraint of Eq. 2 is an inequality constraints of the form $a_i \kappa_i \geq b$, while Eq. 3 demands non-negativity of all κ_i . Together, these constraints define the feasible space as any point κ_i in the positive sector, that exists above a certain hyper-plane (defined by Eq. 2). Therefore, these constraints never exclude all of κ -space. In addition, the optimization functions due to LP and QP generally tend to minimize the different κ_i (e.g., to reduce the energy of the unique mode). We thus generally obtain feasible solutions that saturate the constraints.

2) Reviewer 1 also inquired about why an analogue to equation 2 is not imposed to ensure that the desired deformation mode has lower energy than the transition state. The reply argues that the objective function’s role is to ensure that the desired mode has low energy—while true, this response does not fully answer the question of why inequality constrains were not/could not be imposed. Are such constraints simply redundant? Or are there cases where their inclusion would make the LP infeasible (corresponding to desired modes that cannot be programmed using the paper’s proposed strategy?)

As noted on Pg. 5 and in Supplementary Note 4, the recipe that appears in the manuscript is generically sufficient to obtain stiffness profiles that meet our requirements of lifting all undesired modes.

In principle one could add inequality constraints allowing only stiffness profiles κ_i that lift the desired mode less than the transition state. Such constraints would have the form

$$\frac{1}{2} \rho_c^2 \sum_{i \in \text{creases}} \kappa_i \left[(\tilde{\rho}_D)_i^2 - (\tilde{\rho}_{TS})_i^2 \right] \leq E_{TS}$$

Comparing this to Eq. 2

$$\frac{1}{2} \rho_c^2 \sum_{i \in \text{creases}} \kappa_i \left[(\tilde{\rho}_U)_i^2 - (\tilde{\rho}_{TS})_i^2 \right] \geq E_{TS},$$

it is apparent that the constraints are independent. The extra constraint implies that feasible κ_i 's must be below some hyper-plane. We have sampled over 1000 random vertices and verified that adding the extra constraint never closed off all of the feasible space (i.e. there was always a sizable region in κ -space in which all constraints were satisfied). This however does not preclude the existence of infeasible programs given such constraints. In any case, the recipe that appears in the manuscript is generically sufficient to obtain stiffness profiles that meet our requirements of lifting all undesired modes.

3) Figure 1 is clearer in the new revision, though I still have some suggestions about improving its clarity: - the use of ϕ in this figure (and throughout the main text) is still confusing. Why is $\pi/4$ the transition state, and why are the two configurations shown at the bottom of figure 1a $\phi = 0$ and $\phi = \pi/2$? Could a simpler parameterization of the linkage's configuration be used instead (for instance, the interior angle between one pair of rods) for this figure? - Figure 1c and 1d do not appear to agree: in 1c, the energy minimizer is always at $\phi = 0$, whereas from 1d I would expect curve 1 to have minimizer near $\phi = \phi_{TS}$ (presumably slightly less than $\pi/4$), etc.

We thank the reviewer for the close examination of Figure 1. As discussed in Supplementary Note 1, the use of a mixing angle ϕ in this context is made necessary by the choice of fixing the overall distance from the 'flat state' as $\rho \equiv \|\boldsymbol{\rho}\|$. In terms of describing the energy landscape of the system, this choice is perhaps the most natural. Consequently, taking the two spanning vectors of the linearized null space as the two zero-motions defines them at $\phi = 0, \pi/2$.

The apparent disagreement between Figures 1c and 1d most likely occurs due to the switching the x -axis between the two panels. As detailed in the response to the previous reviewer report, the two perspectives were chosen to illustrate the main idea of the paper to readers from different fields (mechanics and bifurcation analysis). The two panels present the same information in different ways that might appear more familiar to varying audiences.

4) Other minor comments about the text:

- Capital K is not defined (page 2, bottom)
- As discussed in previous reviews, it would be useful to include on page 3 a few sentences about how $\tilde{\phi}_{TS}$ and E_{TS} are computed in practice.

As the reviewer rightly suggested, we now define K on Pg. 2 as the rod compression modulus.

Pg. 3 now contains a sentence clarifying how quantities such as $\tilde{\phi}_{TS}$ and E_{TS} are computed. We now say, "*As the vertex null space (at fixed ρ) is 1-dimensional and compact, these features ($\tilde{\rho}_U$, $\tilde{\rho}_D$, $\tilde{\rho}_{TS}$ and E_{TS}) can all be computed numerically efficiently using peak analysis.*"

- Maybe I'm missing something obvious: why is the RHS of equation 2 E_{TS} , rather than zero?

As explained in the last paragraph of Pg. 3, and shown in figure 2(b), the undesired mode needs to be lifted above the transition state energy. The appearance of E_{TS} on the RHS

of Eq. 2 is meant to guarantee that the undesired mode is lifted more than the transition state. E_{TS} corresponds to the energy of the transition state separating the two modes of the vertex (also the energy difference between the transition state states and undesired mode for $\kappa_i = 0$). If the RHS of Eq. 2 were 0, the constraint could be satisfied for infinitesimally small κ_i 's, which do not lift the energy of any state.

- The paper mentions that, on the one hand, there is typically a large variation in possible folded states when the strain rate is large; on the other hand, the folded pattern in figure 5c is reproducible, even the fast folded state in figure 5c, right. Is there something special about the example in figure 5c that ensures reproducibility? I would expect fast folding to sometimes yield the "slow" or "intermediate" states instead? More discussion would be valuable.

We thank the reviewer for raising the point of folding reproducibility at high strain rates. Once a heterogeneous stiffness profile is established, the flat state branch point is effectively removed, so that the low- ρ behavior is reproducible for any finite strain rate. Now that there are no longer branch points in the folding path, the folding path is reproducible at any strain rate. As such, the example in Figure 5 is generic, and a similar behavior is expected for any pattern in which fast folding evokes a different mode compared to adiabatic folding. To clarify this, we now say "*fast folding from the unique low- ρ minimum reproducibly picks the folded configuration...*"

We thank the reviewer for the considerable effort extended towards detailed understanding and inquiring about both the original manuscript and its revision. These efforts were of great assistance in improving both the content and presentation of our work.